# Multilayer Perceptron for the Future Urban Growth of the Kharj Region in 2040

**Abear Safar Alshahrane** [1] **and Hamad Ahmed Altuwaijri** [2,*]

[1] Department of Geography, King Saud University, Riyadh 11451, Saudi Arabia; abeersafargis@gmail.com
[2] Geography Department, College of Humanities and Social Sciences, King Saud University, Riyadh 11451, Saudi Arabia
[*] Correspondence: haaltuwaijri@ksu.edu.sa

**Abstract:** Urban growth is described as an increase in the size and use of cities, which is frequently the consequence of an increase in the number of residents due to internal or external migration and an increase in economic activity rates. In recent decades, modern technology and mathematical models have been used to determine future urban growth on a large scale and develop sustainable urban policies in the long term. The cities of the Kingdom of Saudi Arabia have witnessed economic growth in recent decades, which has resulted in urban expansion, as is evident in this case study of the Kharj region. Since most of the previous studies have not applied mathematical models to predict the urban growth of the Kharj region, this study aims at simulating urban growth over the next two decades, between 2020 and 2040, by monitoring the growth during the past thirty years, which is the period between 1990 and 2020. This study relies on the satellite visualizations of the Landsat satellites 5, 7, and 8 for classifying the land cover by applying the land change model (LCM) and comparing the land-use maps for the years 2000 and 2020. Then, the factors affecting urban growth, such as distance from the city center, the road network, valleys, and land slopes, are determined to monitor the prediction of urban growth. The results showed that the urban areas extended significantly toward the south, southeast, southwest, and northwest, with an area of 269 km². The results further revealed a significant decline in agricultural and vacant lands due to their transformation into residential areas, educational establishments, and industrial facilities. The model's accuracy was tested to confirm the mathematical model's validity. The Kappa index findings indicated a high percentage, ranging from 89% in 2010 to 90% in 2020.

**Keywords:** urban growth; modeling; GIS; land change modeler; Markov chain

## 1. Introduction

Urban growth is described as an increase in population growth rates, whether this increase results from an increase in population due to internal migration or natural population growth due to economic, social, and educational changes, resulting in urban development for cities [1]. The phenomenon of urban growth, which has been witnessed by many countries around the world, is an indicator of civilizational progress, as 55% of the world's population lives in cities, with that figure expected to rise to 68% by 2050. According to a new United Nations data set, projections indicate that urbanization and the steady shift of population residency from rural to urban regions might add another 2.5 billion people to urban areas by 2050 [2].

The problem of expanding cities is a global phenomenon that is not unique to any one region. However, the urbanization process in the Kingdom of Saudi Arabia is characterized by rapid urban transformation at the expense of cities and villages due to economic and social abundance [3]. According to the United Nations Development Program, the Kingdom of Saudi Arabia's social and economic development has resulted in rapid population growth in many urban areas. The percentage of the population living in urban areas was 58.4%

in 1975, 86.6% in 2001, and 91% in 2016. This rapid population growth has increased the size of the urban area [4]. Among the most prominent of these cities, distinguished by their essential location and rapid population growth, is the region of Kharj. This is because of the availability of services, education, and economic and professional development, which contributed to the urbanization rate, the great demand for land uses in all forms, and the increase in the urban area. The Kharj region is considered one of the regions of cultural settlement and human diversity and one of the most important agricultural regions in the Kingdom of Saudi Arabia [5].

The Kharj region, which includes three small cities (As-sih, Al-hayathem, and Al-Dilam) was, and still is, an attractive area for its population because of the diversity and multiplicity of economic and professional resources available. The emergence of the cities and governorates of the region began in 1939 when people migrated from the old neighboring towns in search of job opportunities. As real urban growth began when the economic boom began and the five-year plans for development in the Kingdom of Saudi Arabia were applied, the urban area in the center of the region, which is the city of As-sih, reached approximately 3.5 km$^2$ in 1974. In 1985, the urban area increased to approximately 12.5 km$^2$, i.e., an increase of 257%. Such an increase continued until the area reached 19.5 km$^2$ in 2000. In 2010, the area reached 39 km$^2$, which is an increase of 100% from the year 2000. This increase was a result of an economic boom and the availability of job opportunities in the industrial, agricultural, and military fields, which in turn led to an increase in urban development in the Kharj region. By 2018, the urban area had expanded by around 60 km$^2$, and this was attributed to the conversion of agricultural lands into residential schemes [5,6], as studies [6,7] highlighted the phenomenal expansion. Thus, by 2010, the region had attained a 35-fold increase in urban area compared to 1973.

Due to a lack of studies on the region, it is necessary to determine the rate of change in land use from 1990 to 2020 to help develop a model for the region's future urban growth for the year 2040.

## 2. Urban Growth Modeling

Urban growth is a dynamic process that changes over time and relies on spatial studies to understand the city's growth pattern, detect changes, and monitor the expansion process, using geographic information systems (GIS) and remote sensing systems (RS). Because it provides information and data in various forms, such as monitoring the process of city growth, understanding urban expansion and trends, and simulating future urban growth by making predictions to direct the city's growth under the given conditions to reduce potential risks, it assists decision-makers and those interested in urban planning in setting plans for urban development in line with the increase in population growth and urban development [8].

Urban modeling is the process of selecting the appropriate theory and rendering it into a formal mathematical model by computer programs related to the model, then comparing the model to the data, calibrating it, and verifying it before using it in prediction [9]. The introduction of urban growth simulation models in Europe and the United States started with the Lowry model in 1946 [10], and further development was undertaken after the 1980s. The methodology followed in the urban growth simulation models has also changed to focus on the details and minute elements after focusing on the more significant elements. It has become an analysis of the factors leading to the pattern of a phenomenon after analyzing the pattern itself. It has considered the spatial and temporal characteristics of the phenomenon [11].

Current trends in urban growth indicate that urban areas are expanding, increasingly and rapidly encompassing vast rural areas, and that major cities are dominating the general urban landscape and have become one of the main and distinctive features of many Arab countries in recent years. This situation, which characterizes urban development in the world, creates great difficulties in the development process, as the issue of not keeping pace with an appropriate transformation in the social and economic structure is raised, which

causes many different problems and resulting socio-economic issues. When formulating development policies that seek to plan for the phenomenon of urbanization to confront its consequences and control its movements, urban growth requires an approach in the areas of public policy, urban planning and management, and the development of more sustainable strategies [12].

## 3. Literature Review

Many studies have applied urban growth simulation models to study land-use change and predict future urban expansion, such as the CA-Markov chain model, the land change model (LCM), and the logistic regression model, by representing them in various GIS programs [13,14]. A city's urban change was disclosed throughout the past 30 years, from 1987 to 2017, through a study predicting Riyadh's urban growth. After analyzing the data, the geographical factors affecting the city's future urbanization were introduced. The study reached several conclusions, most notably that the expected urban expansion for Riyadh was about 27% [14].

Another study [15] created a model that was calibrated using historical digital maps of urban areas. The model was tested in two rapidly growing areas: California's San Francisco Bay Area and Washington. Differences in urban expansions and compositions were compared, and growth rates were noted. The studies recommended using urban expansion studies and future growth predictions to support decision-makers in urban planning [14,15].

Al-Darwish [16] also studied the prediction of future urban growth and its impact on the Yemeni city of Ibb using a Markov cellular model integrated with a GIS platform and model validation to simulate urban growth for the year 2033. The results showed the accuracy of the prediction in reaching the horizontal distribution of urban growth at a high rate at the expense of agricultural and natural lands. The study also suggested that decision-makers reevaluate their expansion strategies to maintain ecological balance.

Another study [17] also recommended this, using the Markov model and multi-layer perceptron (MLP) to detect changes in land cover in the municipality of Pabna in Bangladesh between 2023 and 2028. The results showed that the urban area increased from 3.39 km$^2$ to 8.79 km$^2$, and the direction is mainly toward the northeast, meaning that the urban construction land area will grow to 11.51 km$^2$ in 2023, and 12.44 km$^2$ in 2028, heading toward the northeastern part. That will assist urban planners in determining urban growth and preparing appropriate strategic measures.

On the other hand, some studies used more than one model to predict urban growth; for example, a study of the town of Sakib relied on applying the most prominent environmental simulation models available in the IDRISI Selva (Version 17.01) software. Through this, it was possible to predict the urban development of the town of Sakib until 2040. The results of land change modeling and Markov cellular modeling showed the development of urban expansion over an area of 2.56 km$^2$. Further, there were differences between the two models in the spatial distribution of urban growth from site to site [13].

Jokar Arsanjani [18] applied the logistic regression model and the Markov chain model, making an interchangeable comparison between the simulation maps and the actual maps for the year 2006 and then predicting the future land-use maps for the years 2016–2026. The study showed a new wave of suburban development along the western borders of Tehran in the coming decades.

Rapid urbanization rates also harm a city's development and demographic factors, leading to unplanned and unsystematic growth and slowing down the process of urban sustainability, among other things [19]. Therefore, urban growth models provide a better understanding of spatial and temporal patterns and predict future changes, carefully choosing the simulation model that best matches the changes in the study area [20].

Based on the above presentation of previous studies, it is clear that urban studies have received attention from scientists and planners, applying mathematical models to simulate the future urban growth of most cities according to their circumstances, geograph-

ical locations, and the places on which they were based. This study emerges from this framework, but the difference is that satellite visuals were used for previous years and compared to reality for 2010, depending on the period between 1990 and 2000 and also the year 2020, depending on the period between 2000 and 2010, to verify the accuracy of the earth change model.

## 4. Area of Study and Study Methods

### 4.1. Study Area

The Kharj region (Figure 1) is located in the center of Saudi Arabia, within the continental shelf, southeast of Riyadh. It extends between latitudes 23.10 and 24.30 north and longitudes 46.30 and 48.15 east [21], with an elevation of roughly 360 feet above sea level. The region also controls the central basin of Wadi Sahba, which contains several valleys. It is considered one of the most prominent regions in the Kingdom in terms of plant diversity due to the dominance of the steppe environment over most of the regions [5].

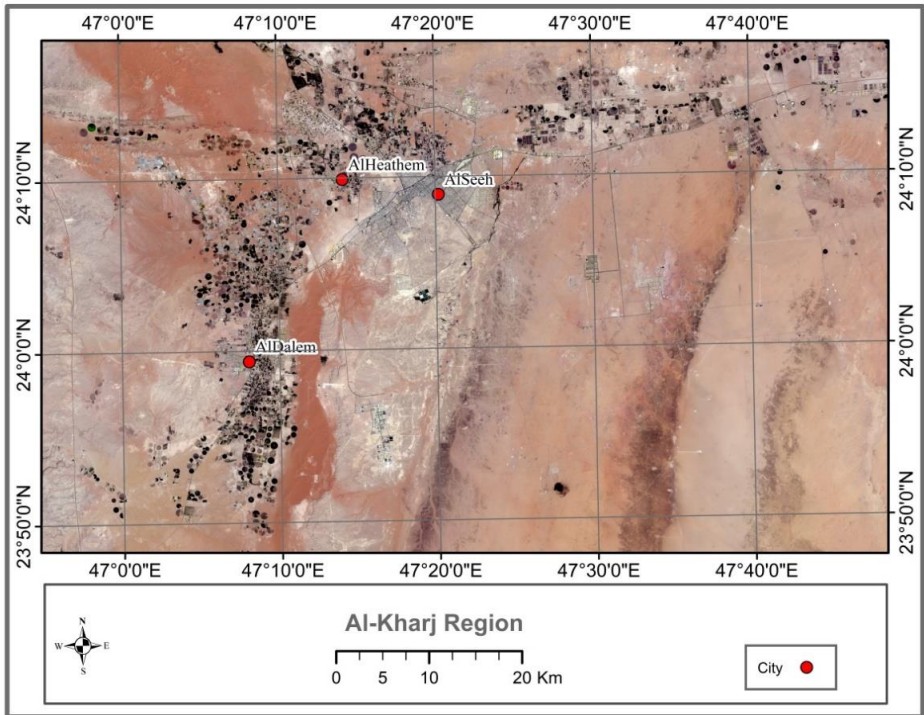

**Figure 1.** Satellite visualization of the Kharj Region. Source: United States Geological Survey (USGS).

### 4.2. Study Approach

Based on the nature of the study and its objectives, it relied on descriptive and analytical approaches. A descriptive approach means describing the phenomenon as it exists in reality. An analytical approach means the method used to collect information about a water phenomenon or a specific reality in order to identify the current situation [22]. Moreover, the methodological limitations of this study include factors that affect urban growth, such as distance from the city center, road network, valleys, and land slopes.

#### 4.2.1. Study Methodology

The procedures of the study and its most essential phases start from collecting data, classification of Images, building a land change modeler, and all factors that affect the growth to come up with prediction maps are shown in the following Figure 2.

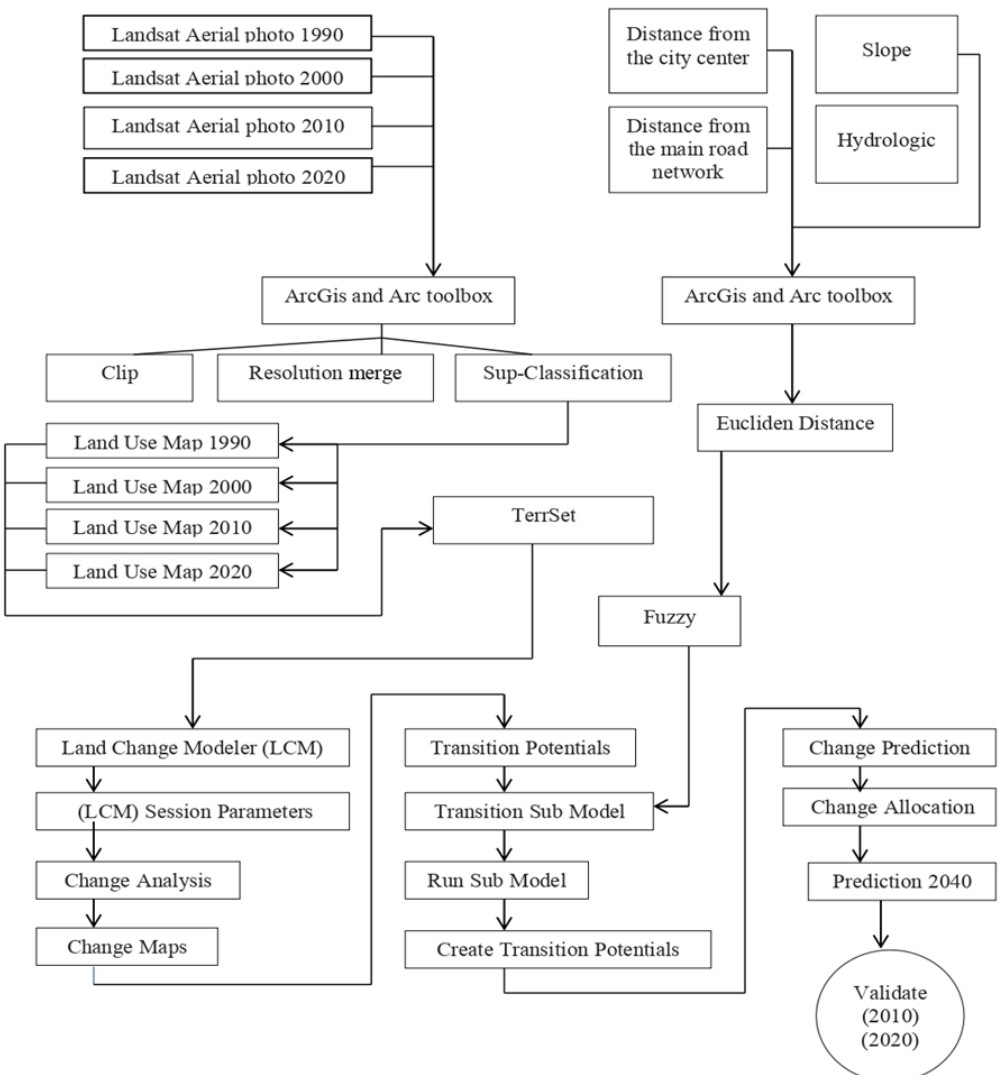

**Figure 2.** Study Methodology.

4.2.2. Data Preparation

The study relied on four satellite visualizations to track the evolution of land cover areas in the region from the beginning of 1990 to the end of 2020. It was also taken into account that all visualizations were captured simultaneously to achieve equality at the regional level without the effect of clouds. The Universal Transverse Mercator (UTM) projection was used since the projection used for all satellite visualizations was unified. This approach places the research area in zone 36 N, with reference to WGS 84. Enhancements were performed to improve the degree of clarity (spatial accuracy) of the resolution merge, and then conduct the monitored classification of satellite visualizations for the study years 1990, 2000, 2010, and 2020 into three main categories: vacant lands, built-up lands, and agricultural areas.

4.2.3. Modeling Future Urban Growth

In this study, the land change modeler (LCM) was used to generate transition matrix maps using the Markov algorithm and the multilayer perceptron (MLP). The study's data and visuals were structured by converting them from Raster to ASCII format to get the desired outcomes of applying the model. It was also necessary to organize the data to avoid errors and take into account the similarity of the map keys and the classification

categories. Likewise, the images must be similar in terms of spatial dimension, geographical coordinates, and spatial clarity.

## 5. Analysis and Results

### 5.1. Urban Growth of the Kharj Region

Based on the tremendous urban growth witnessed by the Kharj region [5], it has become necessary to understand and study urban development and its trends and interpret that growth to direct future development and urban growth. The urban growth areas for the research groups were calculated using the controlled classification into vacant lands, built-up lands, and agricultural areas, as shown in Table 1.

**Table 1.** Land cover areas of the Al-Kharj region during the study period.

| Years | Area in km$^2$ | | |
| | Vacant Lands | Built-Up Lands | Agricultural Areas |
|---|---|---|---|
| 1990 | 2391 | 47 | 183 |
| 2000 | 2366 | 71 | 184 |
| 2010 | 2290 | 117 | 214 |
| 2020 | 2315 | 161 | 145 |

We discovered that the period from 1990 to 2000 experienced economic and social changes after the economic boom [5], which significantly impacted the Kharj region's urban growth because the area of urban spaces in 2000 reached 71 km$^2$ at a rate of 2.7%. It was 47 km$^2$ in 1990, representing a 24 km$^2$ increase. It extended along the agricultural areas because of the presence of valleys and wells that feed the agricultural areas, making up 7% of the overall land cover and covering an area of 184 km$^2$ during the year 2000. At the same time, vacant land constituted the most significant percentage, ranging between 90% and 91% of the total area during the ten years.

In 2010, the area of urban spaces expanded by 65%, from 71 km$^2$ to 117 km$^2$. It began to extend along the main roads of As-sih, Al-hayathem, and Al-Dilam, as well as surrounding agricultural areas, accounting for 8.16% of the Kharj region's land cover area of 214 km$^2$. The vacant land area decreased by 3% in favor of residential areas, as it extended in the eastern and southwestern directions. Urban growth began to spread widely in urban areas in 2020, with an area of 161 km$^2$ and a rate of 6.14%. That is a result of the demand for residential land and the conversion of agricultural areas into vacant lands, with a 69% area conversion. The decline in agricultural areas is about 131 km$^2$ of land cover (see Table 2 and Figures 3–5). In addition, as shown in Figure 5, there are new growth areas that have appeared 15 km southwest of the intersection of Al-Dilam Road and the road connecting to Al-Kharj Industrial City. As part of industrial regional planning, these new growth areas were established in 2011 and are still expanding as industrial areas.

**Table 2.** The difference in land cover areas.

| Urban Cover | The Difference in Areas in km$^2$ | | |
| | 1990–2000 | 2000–2010 | 2010–2020 |
|---|---|---|---|
| Vacant lands | 25− | 77− | 26− |
| Built-up lands | 24+ | 46+ | 44+ |
| Agricultural areas | 1+ | 30+ | 69− |

Source: the researcher's work, based on classifying the visuals of the years of study.

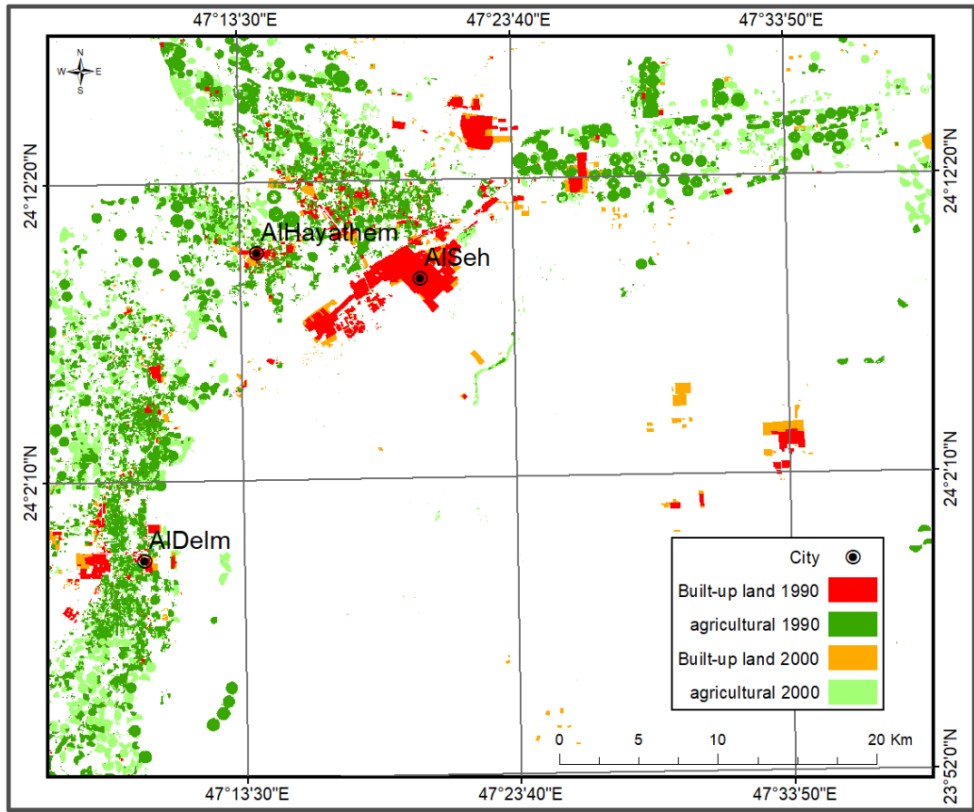

**Figure 3.** The urban growth of the Kharj Region during the period 1990–2000.

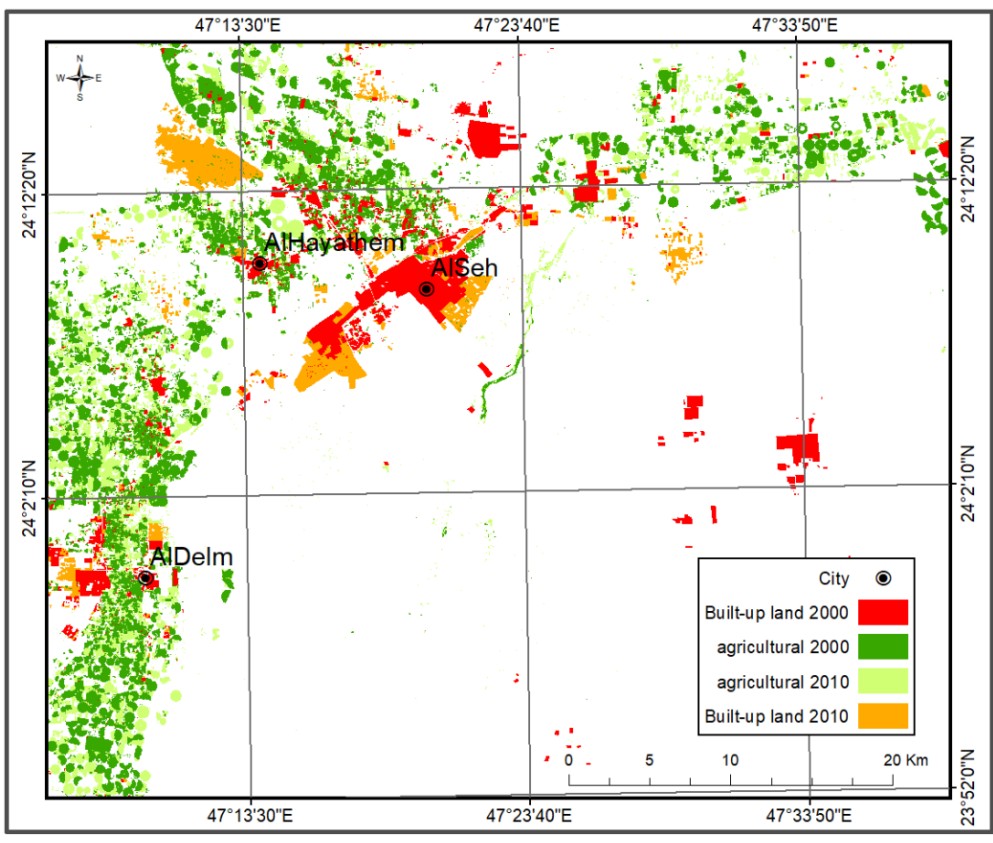

**Figure 4.** The urban growth of the Kharj Region during the period 2000–2010.

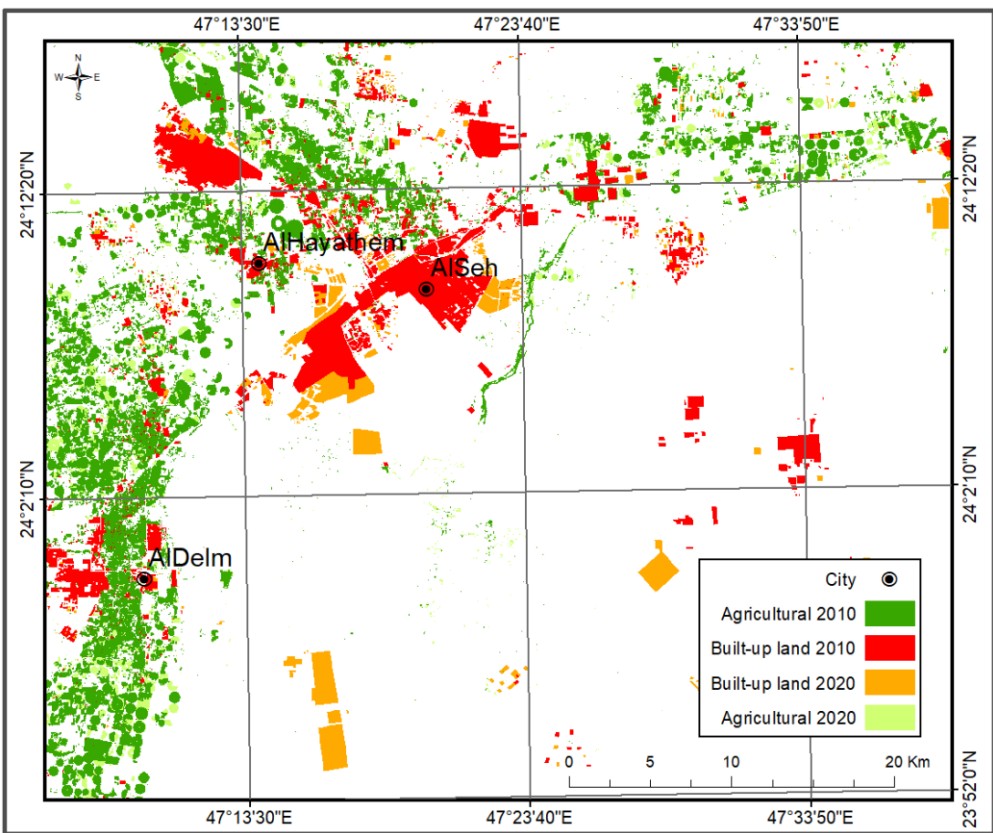

**Figure 5.** The urban growth of the Kharj region during the period 2010–2020.

*5.2. Checking the Accuracy of the Land Change Modeler (LCM)*

Multilayer perceptron (MLP) was used to obtain transition matrix maps based on the Markov method to identify transition and project change zones to predict the years 2010 and 2020. The maps were obtained based on a literature study of prior works interested in applying mathematical models to simulate urban growth. The IDRISI Terrset (v19.0.6) software's statistical geographic information systems analysis tools were used to compare the simulation maps' (predicted) results, with a base map to validate the model's accuracy by Kappa statistical. Cohen's Kappa coefficient was developed in 1960 to measure reliability between two raters [23]. The formula is as follows:

$$k = (Po - Pe)/(1 - Pe)$$

where:

Po: Observed agreement among raters;

Pe: Hypothetical probability of chance agreement.

It was used in 1980 in remote sensing to express the accuracy of an image classification [24], which can be expressed in remote sensing as,

Kappa = (total accuracy – random accuracy)/(1 − random accuracy). The validation was carried out for the following years:

Verification for the year 2010: The general Kappa statistic for simulating land change modeling reached 0.89 when comparing the 2010 map predicted by the analysis of the two visuals for the years 1990–2000 with the classified 2010 map. That indicates the ability of the model to simulate in a manner that excellently serves the study (See Figure 6).

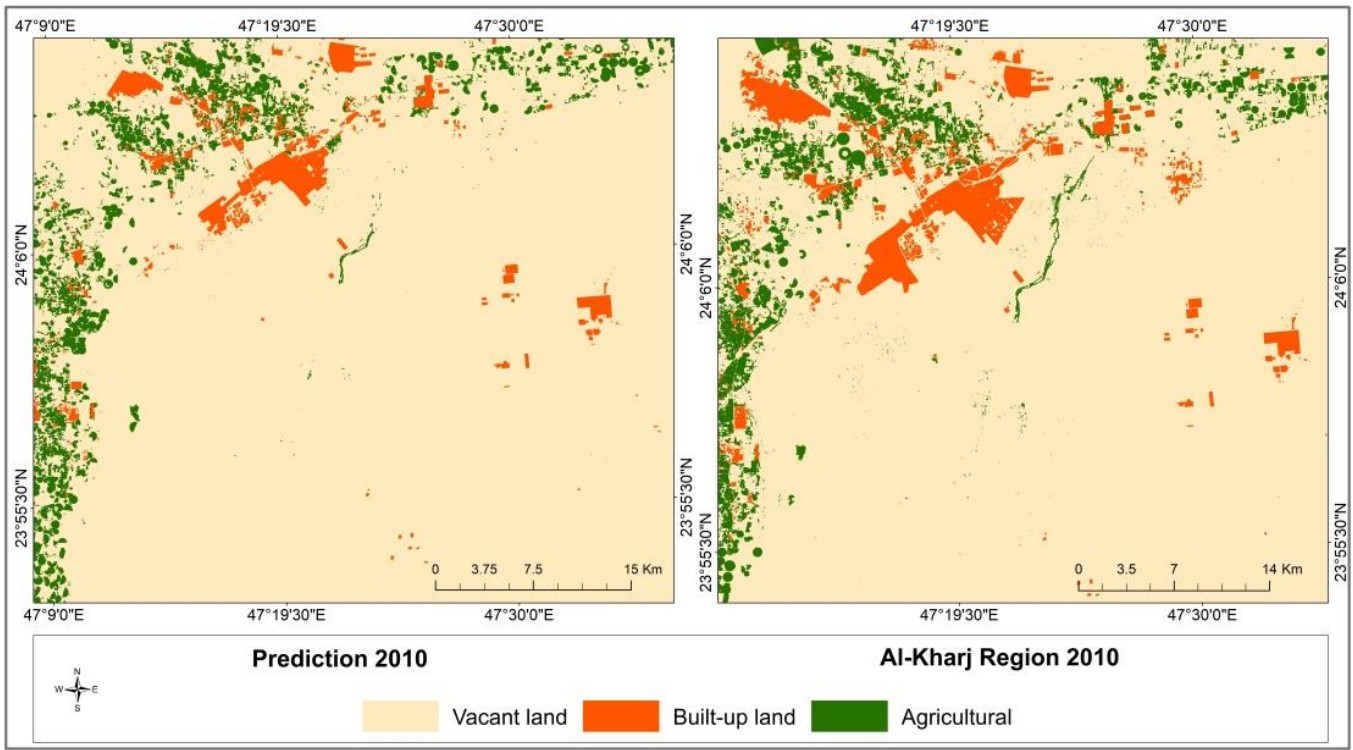

**Figure 6.** Verification for the year 2010.

Verification for the year 2020: In comparing the 2020 map predicted by the analysis of the two visualizations for the years 2000–2010 with the classified 2020 map, where the Kappa value = 0.90 indicates a high accuracy in the simulation process, and the cells of each of the base map and the simulation map are correct (see Figure 7 and Table 3).

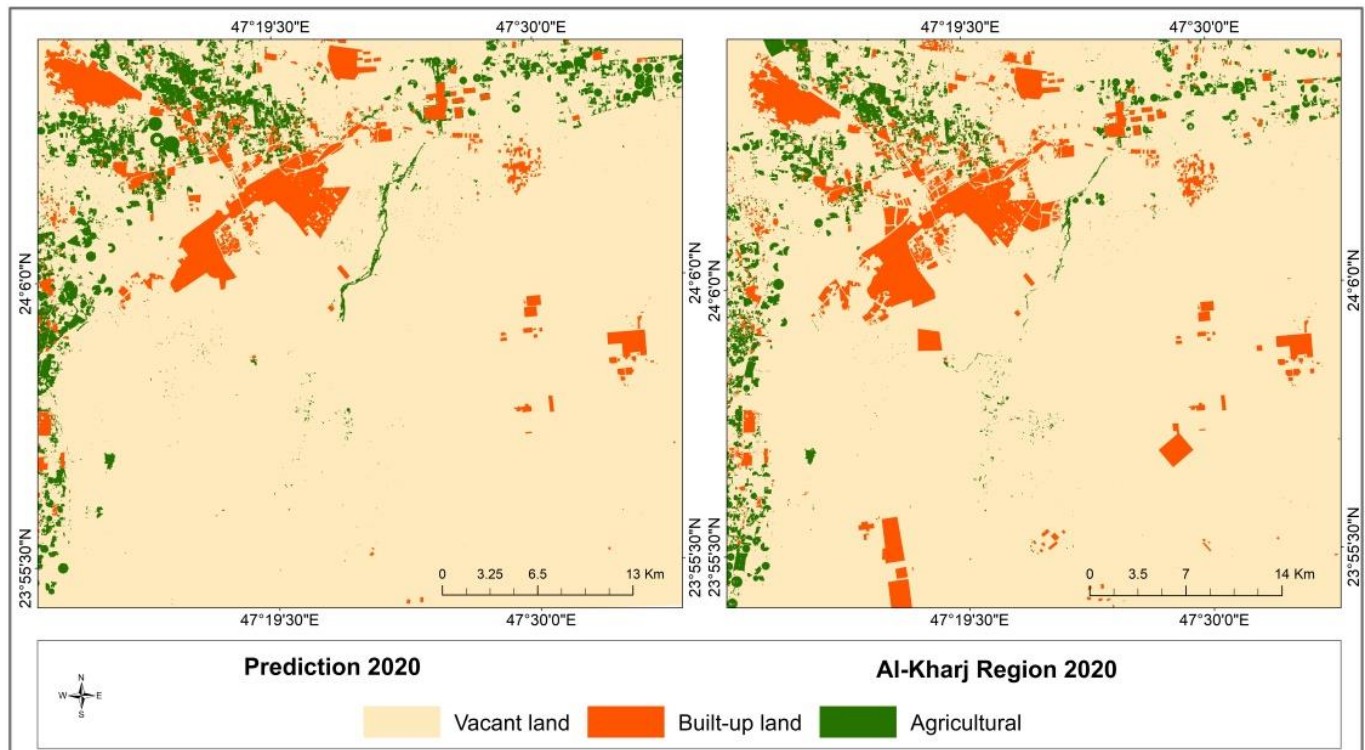

**Figure 7.** Verification for the year 2011.

**Table 3.** Kappa index statistics.

| Kappa Index | Land Change Modeling (LCM) for 2010 | Land Change Modeling (LCM) for 2020 |
|:-----------:|:-----------------------------------:|:-----------------------------------:|
| Kno | 0.89 | 0.90 |

### 5.3. Modeling Future Urban Growth for the Year 2040

The land change model (LCM) was used in the study to analyze and compute the amount of change between the land-use map from 2000 (Figure 8) and the land-use map from 2020 (Figure 9). The study used spatial modeling to predict changes in land cover.

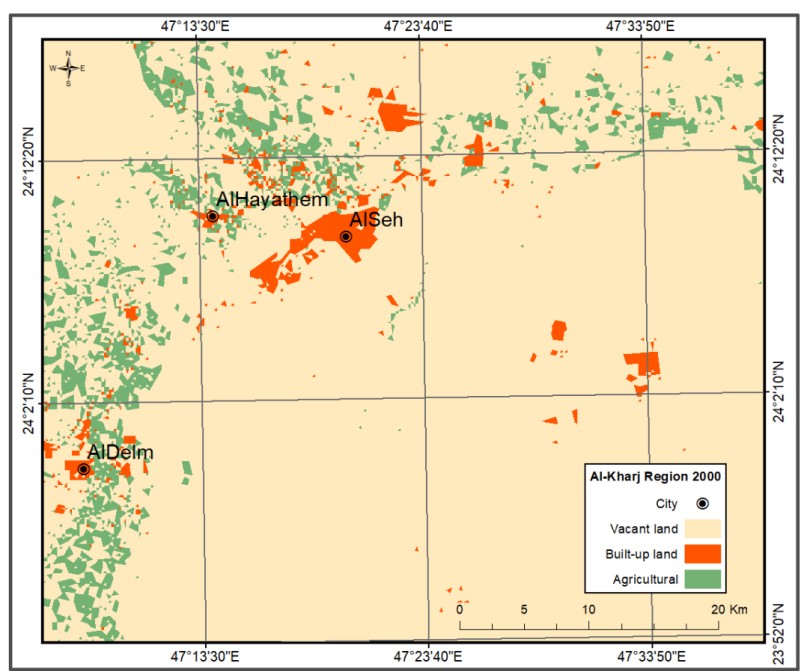

**Figure 8.** Land-use map for the year 2000.

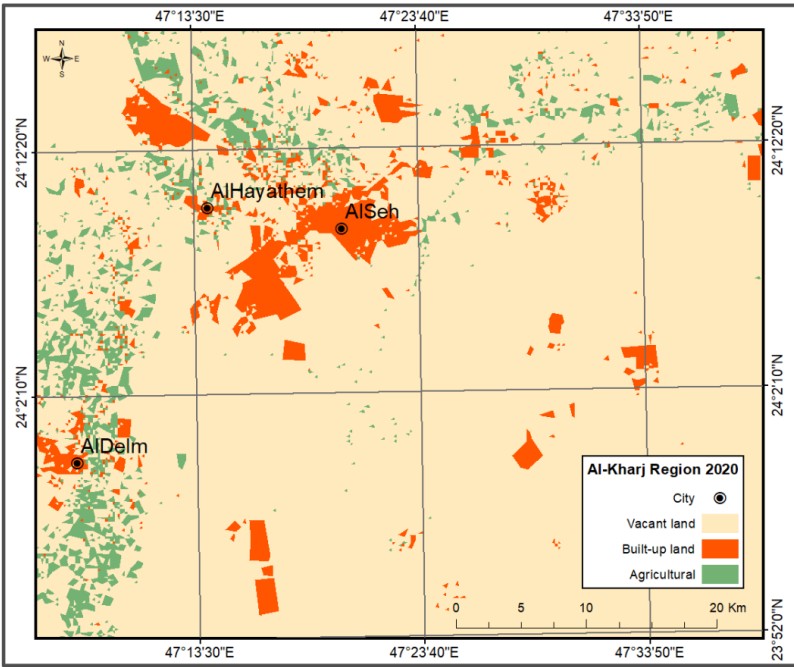

**Figure 9.** Land-use map for the year 2020.

5.3.1. Change Analysis

A name was given to the project through the project data window, and three different land-use maps (agricultural areas, built-up areas, and vacant lands) were entered to carry out the analysis.

Change analysis was the first step, providing a quick assessment of the quantitative change through a graph of the gains (green) and losses (purple) for the various land cover categories per kilometer (See Figure 10).

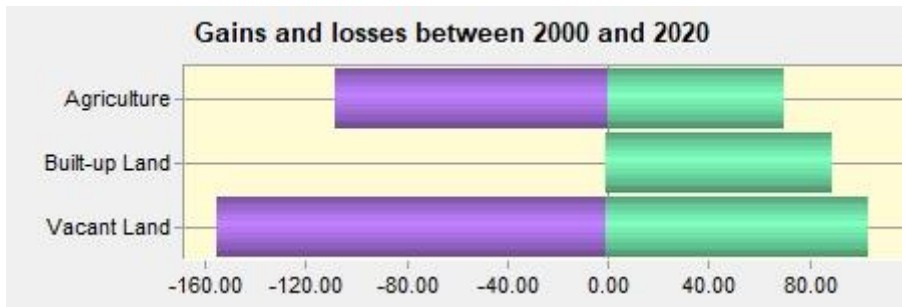

**Figure 10.** Gains and losses in the areas of the categories in km$^2$.

We found that over the study period, the area's vacant lands shrank by 155 km$^2$, whereas other regions saw 104 km$^2$ growth, resulting in a loss of vacant lands at a pace of 51 km$^2$. On the other hand, the rise in built-up land—which amounted to nearly 90 km$^2$—is the predominant change between 2000 and 2020, and the area's loss is essentially nonexistent. The decline in agricultural areas became prevalent in many regions as the cultivated area decreased by approximately 108 km$^2$. In comparison, it increased in other regions by approximately 70 km$^2$, indicating that the trend here was downward (See Figure 11).

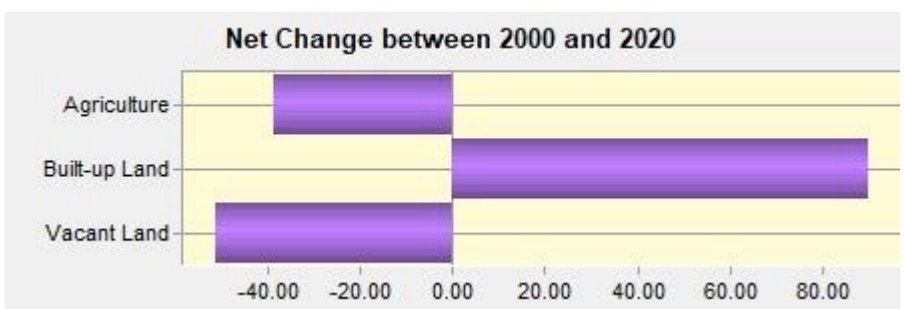

**Figure 11.** Net change of areas in km$^2$.

We found that the net area is the difference between gains and losses, which supports the previous conclusion that built-up land areas increased by about 90 km$^2$ over the study period, while land areas decreased by 51 km$^2$ and agricultural areas decreased by 39 km$^2$. Additionally, each category's percentage change is computed using the following formula:

$$\text{Percentage net change in area} = \left[ \frac{(\text{new area} - \text{original area}) \text{of the category}}{\text{new Area of category}} \right] \times 100$$

The pace of growth in urban areas was 55.9% of the total built-up land area. The percentage of the decrease in the area of vacant lands took up 2.2% of the area of the studied category. Regarding the agricultural areas, cultivated areas decreased by roughly 27% as a percentage (Figure 11 and Table 1).

In order to determine the spatial distribution of the changes between the categories and determine their areas, a change map relating to the changes in the classified categories for the years 2000 to 2020 was also developed (see Figure 12 and Table 4).

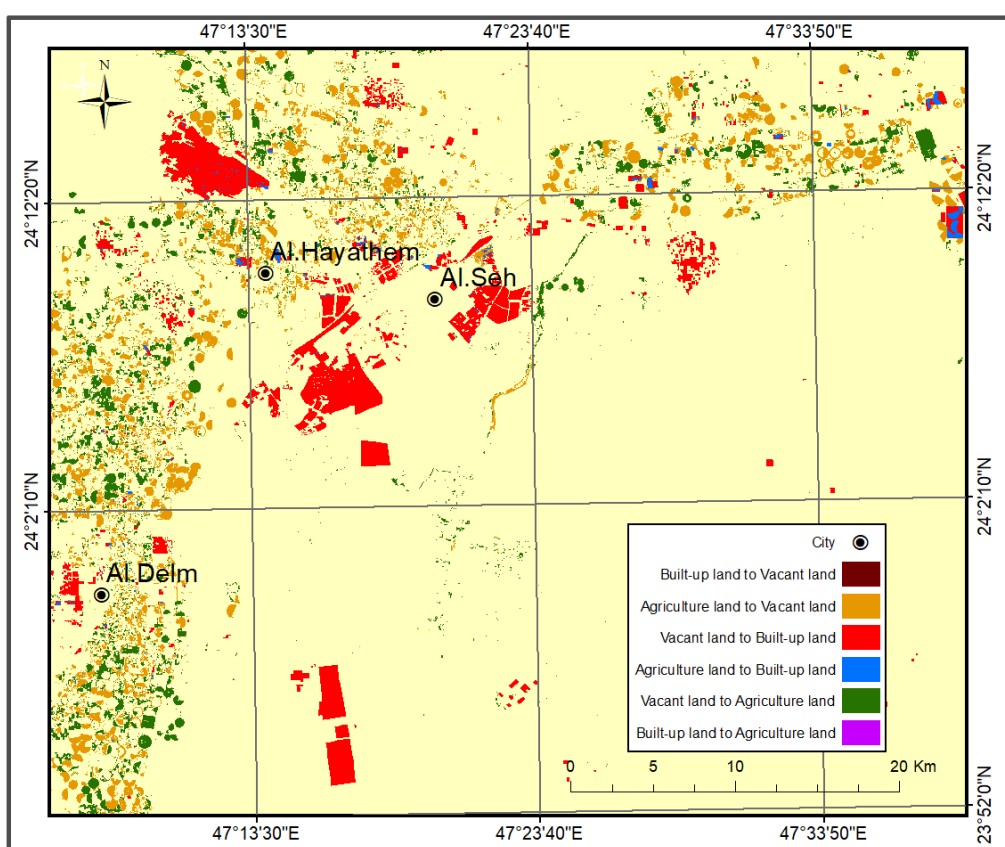

**Figure 12.** A map of the changes that occurred for the studied categories between the years 2000 and 2020.

**Table 4.** Areas of change among the studied categories between the years 2000 and 2020.

| Order | Area in km$^2$ | Category |
|---|---|---|
| 1 | 0.11 | Built-up lands to vacant lands |
| 2 | 104 | Agricultural areas to vacant lands |
| 3 | 85 | Vacant lands to built-up lands |
| 4 | 4.6 | Agricultural areas to built-up lands |
| 5 | 69.9 | Vacant lands to agricultural areas |
| 6 | 0.4 | Built-up lands to agricultural areas |
| 7 | 2357 | Unchanged areas |
| Total | 2621 | Total area |

Based on the spatial trend of change analysis tool for the classification variables, it became clear that the urban extension of the Kharj region was toward the north, south, and southeast. The As-sih governorate extended toward the east, along the main roads of the three provinces of the region As-sih, Al-hayathem, and Al-Dilam. Moreover, the extension of the region receded to the west because of the dunes (Figure 13).

5.3.2. Transition Potential

The function of this stage lies in creating potential transition maps to run the actual modeling and then assembling transitions into a set of sub-models and exploring the latent power of explanatory variables [25]. Six transitional groups were identified for the categories or classification variables of urbanization, farms, and vacant lands and included in one group under the name "To Urban" to create transition probability maps.

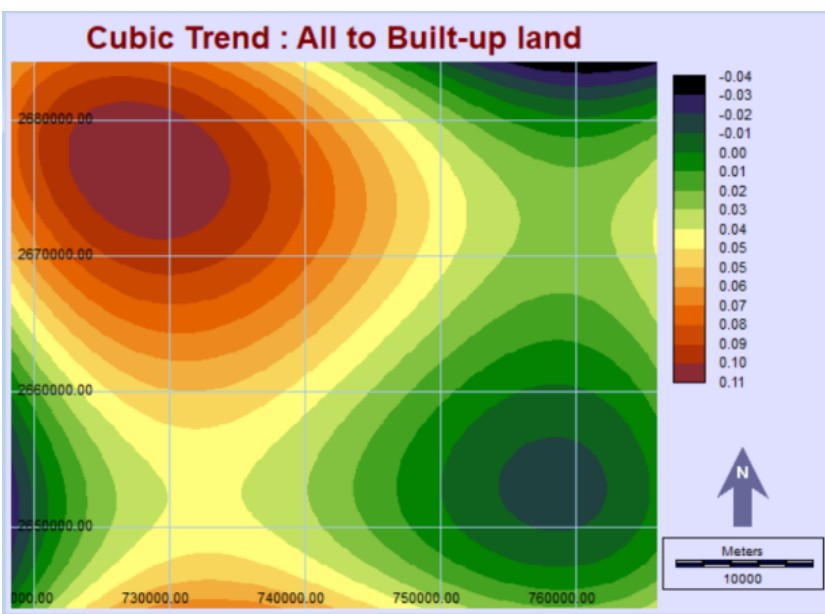

**Figure 13.** The spatial trend of urban growth.

Based on a set of criteria influencing the urban growth process of the Kharj region through the transition sub-model panel, a set of criteria was adopted that played a dynamic role in the study area. It was reclassified on a standard scale = (0, 2.55) by making it fuzzy within the geographic information systems analysis tools for decision support within the IDRISI Terrset software.

It is optional to carry out the calibration process for the factors, as the criteria can maintain their original values of distances and heights without being standardzed [26]. The most important of these criteria are:

1.  Distance from the road network (Figure 14);
2.  Distance from city centers (Figure 15);
3.  Slopes (Figure 16);
4.  Hydrology (Figure 17).

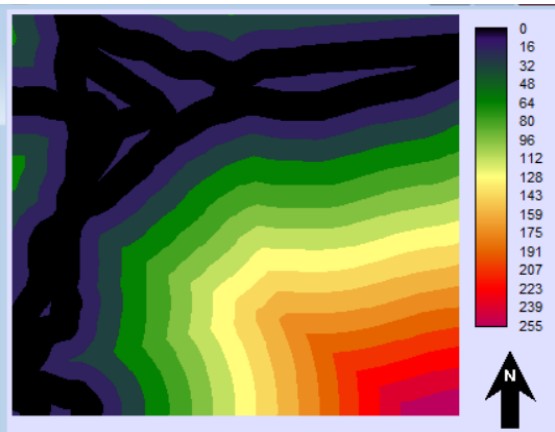

**Figure 14.** Distance from the road network.

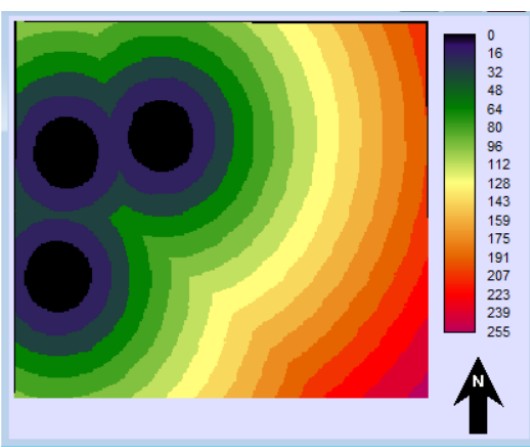

**Figure 15.** Distance from city centers.

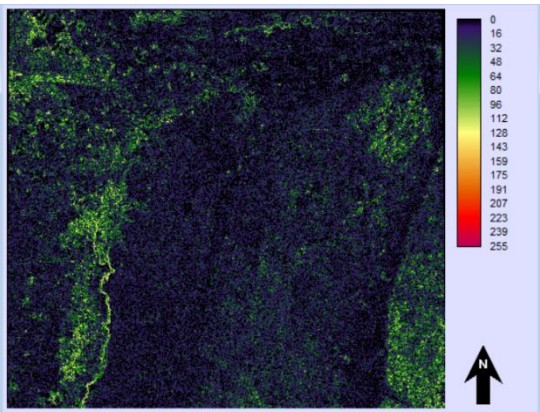

**Figure 16.** Slopes.

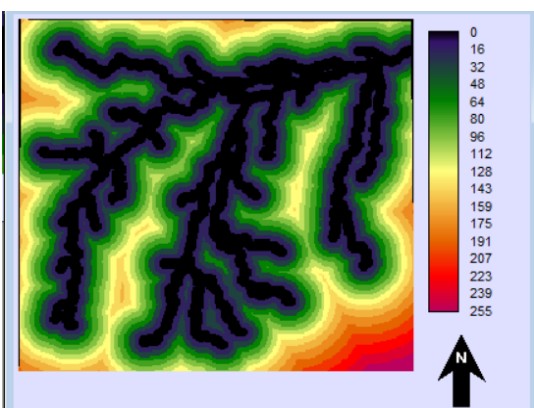

**Figure 17.** Hydrology.

After defining the criteria, the transitional modeling process was implemented according to the methodology of one of the types of artificial intelligence, the multilayer perceptron (MLP). It is one of the most common artificial neuron networks that depend on the BP algorithm [27], as it supports several changes for the studied groups. It was used to extract training samples from the land use maps for 2000 and 2020 that show the areas that have transformed and the areas that have stayed the same in preparation for the machine learning process that will predict future changes.

### 5.3.3. Change Prediction

The urban growth of the Kharj region was predicted for the year 2040 through Markov chain modeling, which is available within the change prediction phase (Figure 18). The amount of land cover used in the previous years (2020) was determined and anticipated for the following years based on that (2040). The transition matrix is derived using future projections and a file of transition possibilities, which can be considered a matrix in which the probability of changing each land cover category is recorded (Table 5).

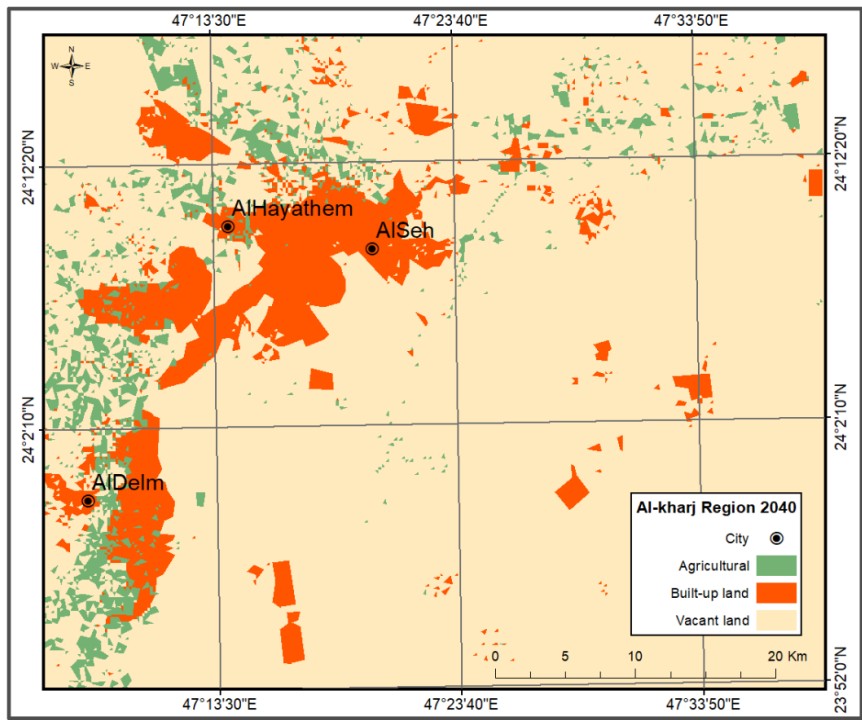

**Figure 18.** Prediction map for the Kharj region in 2040.

**Table 5.** Transition matrix 2020.

| Land Cover | Vacant Lands | Built-Up Lands | Agricultural Areas |
|---|---|---|---|
| Vacant lands | 0.9344 | 0.0360 | 0.0296 |
| Built-up lands | 0.02 | 0.9998 | 0 |
| Agricultural areas | 0.5675 | 0.0251 | 0.4075 |

Figure 18 illustrates the comprehensive and rapid expansion of urban areas. Table 6 shows that urban areas covered about 269 km$^2$, which is 67.1% more than in 2020 when it was only 161 km$^2$. The Kharj region began to expand in several directions as the built-up lands extended in the northwest direction and to the southeast of the city of Al-hayathem. It also extended longitudinally toward the east, west, and southwest of the city of As-sih, along the main roads of Al-hayathem and As-sih. The built-up lands east of Al-Dilam stretched from north to south, paralleling the main road (Riyadh-Wadi AlDawasir).

Estimates show that residential plans will take up 5% of the total land cover area by 2040, which will cause a significant drop in agricultural land. The vacant lands also retreated from what they were previously, amounting to 2220 km$^2$. This is due to the transformation of many agricultural areas and vacant lands into residential developments and educational and industrial facilities. Field research showed an industrial area in the region's southeast; urban growth began to extend near it, and the dunes began to recede for the urban development process. It also became clear that the process of urban growth was concentrated near the city centers and the leading road network.

**Table 6.** The land cover areas of the Kharj region between 2020 and 2040.

| Land Cover | Area in km$^2$ | | % | Difference in Area (in km$^2$) |
| --- | --- | --- | --- | --- |
| | 2020 | 2040 | | |
| Vacant lands | 2315 | 2220 | 85 | 95− |
| Built-up lands | 161 | 269 | 10 | 180+ |
| Agricultural areas | 145 | 132 | 5 | 13− |
| Total | 2621 | 2621 | 100 | |

Source: the researcher's work.

## 6. Discussion

Based on the analytical process of land-use maps for the Kharj region, the study showed that there had been changes in the land cover during the past thirty years (1990–2020). Built-up land areas reached 47 km$^2$ in 1990 and increased to 2.71% of the studied area to 71 km$^2$ in 2000. This increase is slight due to the economic recession that accompanied the drop in oil prices and the Gulf crisis [5], and it continued to increase in 2010, becoming 4.5% of the studied area. In 2020, it became 161 km$^2$, or 6.14% of the studied area, meaning that the Kharj is witnessing a noticeable decline in agricultural lands at the expense of built-up lands. This decline is primarily due to the transformation of agricultural areas into commercial and economic areas, in addition to residential expansions and natural causes, such as sand encroachment and the presence of valleys that formed the urban growth paths of cities [28].

On the other hand, the prediction accuracy was verified twice using the Validate tool, first by checking the visualization of 2010 and then of 2020. The verification rate was excellent, and the value of the Kappa index exceeded 90% This shows the accuracy and importance of the model in urban studies through the drawing up of future policies for planning and sustainable development. The land-use map for the year 2040 also showed that there would be much more built-up land to the south, southeast, southwest, and northwest, with an estimated 269 km$^2$. This result is in line with a population growth study of the Kingdom of Saudi Arabia, which indicates that the population will continue to increase until the year 2060 [29]. Agricultural lands would also lose much land area to residential plans, making up about 5% of the total land cover area. This result may affect biodiversity, as Ayasreh [13] indicated in his study of the Saqib region that the decrease in agricultural areas may be due to the depletion of farms, which drives an imbalance in biodiversity.

Additionally, the area of vacant land had shrunk from its prior size of 2220 km$^2$. That is due to the transformation of many agricultural areas and vacant lands into residential developments and educational and industrial facilities. Accordingly, the land change model (LCM) has been proven to effectively predict urban growth to keep pace with the continuous updates of the urban mass, which gives authority to planners and those interested in setting the expected scenario for urban development and achieving sustainability [30].

## 7. Conclusions and Recommendations

In this paper, urban growth was calculated using historical data and the MLP method. Moreover, the limitations of factors that affect urban growth were considered necessary to produce accurate results. After validation has been applied to this methodology, the researchers recommend using mathematical models as a powerful and effective tool for simulating future growth to understand the expected urban scenario in light of the fast growth of the Kharj region. That helps the city grow in a way that is sustainable and fits with the Kingdom of Saudi Arabia's vision for 2030.

More future forecast studies should be conducted on the impact of urban expansion at the expense of agricultural areas to devise development plans that enhance environmental life and landscapes and develop green spaces to achieve the Saudi Green Initiative.

For future studies, it is recommended to study the integration between the logistic regression model and the Markov chain to simulate the urban expansion of the Kharj region.

The future urban growth of the cities of the Kharj region can also be analyzed using the surface model algorithm.

**Author Contributions:** Conceptualization, H.A.A.; Validation, A.S.A.; Formal analysis, A.S.A.; Supervision, H.A.A. All authors have read and agreed to the published version of the manuscript.

**Funding:** The authors extend their appreciation to the Deputyship for Research & Innovation, Ministry of Education in Saudi Arabia for funding this research work through the project no. (IFKSURG-2-519).

**Institutional Review Board Statement:** Not applicable.

**Informed Consent Statement:** Not applicable.

**Data Availability Statement:** Not applicable.

**Conflicts of Interest:** The authors declare no conflict of interest.

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
