# Peer review of "Multilayer Perceptron for the Future Urban Growth of the Kharj Region in 2040"

_sustainability, doi:10.3390/su15097037_

Round 1

Reviewer 1 Report

The topic discussed in this article is very interesting. It describes the use of various models in detecting changes in landuse in Kharj Region. 

However, it would be good if the authors can enhance the discussion section by relating the findings of this article with the previous studies using the same model and other models. 

Is there any similarities or differences between areas and models.

Author Response

However, it would be good if the authors can enhance the discussion section by relating the findings of this article with the previous studies using the same model and other models. 

Yes, we added two more studies in the discussion section

Is there any similarities or differences between areas and models.

We added a section at the end of the literature review to explain this

Reviewer 2 Report

Thanks to the Authors for the important study related to urban growth modelling.

-The literature is limited in the article. It is important to explain recent trends in urban growth model approaches.  Why land use change modeler is important and why did the authors select this model? The advantages and disadvantages should be mentioned. 

-The case area urban growth modelling and analysis is important forlocal researchers and planners or decision-makers. However, it is not for international scholars. 

-The land use change model may be used with some planning or urban growth scenario for interdisciplinary approach. This may improve the interest for international scholars.  

-Since the journal related to the sustainability, land use change modeler can be used for sustainable urban growth approach. 

Author Response

Thanks to the Authors for the important study related to urban growth modelling.

-The literature is limited in the article. It is important to explain recent trends in urban growth model approaches.  Why land use change modeler is important and why did the authors select this model? The advantages and disadvantages should be mentioned. 

We added more literature

.  Why land use change modeler is important and why did the authors select this model?

We add some sentences in the discussion section to support this query

-The case area urban growth modelling and analysis is important forlocal researchers and planners or decision-makers. However, it is not for international scholars. 

Agree the location is not for international scholars but could be interesting to the government sector in Saudi Arabia. Moreover, this methodology and verifying the prediction accuracy is important for international scholars. 

-The land use change model may be used with some planning or urban growth scenario for interdisciplinary approach. This may improve the interest for international scholars.  

Yes agree

-Since the journal related to the sustainability, land use change modeler can be used for sustainable urban growth approach. 

Thanks, we did sentence support this, in discussion section

Reviewer 3 Report

General overview of the paper and major issues for review:

The paper sets out to simulate urban growth in Kharj over the 2020-2040, by modelling it based on the growth observed in the region over the period between 1990 and 2020.  There is nothing mentioned in that paper to suggest that changing circumstances within KSA, and worldwide, are taken into account in setting up the model.

The authors acknowledge that urban growth is driven by “internal or external migration and an increase in economic activity rates”, but there is no information on where predictions of population growth, birth and migration rates and possible changes in social and economic planning are taken into account.  Given the fast changing circumstances in Saudi society, these would not be expected to remain constant at rates given in the 1990-2020 period.  

Urban growth in terms of land usage would also be determined by planned housing style and density, factors which one would expect will change over time, particularly if the current Saudi policy focus on sustainability within cities and responses to energy concerns, extend beyond the new cities under development.  

The growth forecast in the paper appears to consider only the physical characteristics of the area as determining factors, with the authors listing “the factors affecting urban growth, such as distance from the city center, the road network, valleys, and land slopes, to monitor the prediction of urban growth.”  Social factors (such as future economic activity rates) although briefly acknowledged as important, do not appear to be modelled.  Neither is there any mention of wider social or political policies that will impinge on growth.  It is not realistic to assume these will be unchanged over time.

While the model may suggest the likely direction of urban growth in the area over time if the area expands (based on physical characteristics of the terrain), it seems unrealistic to suggest the speed of growth can be accurately predicted in a context void of the factors highlighted above.

If the wider social and economic context has been taken into account in modelling, this needs to be made explicit by the authors. If it has not been, or cannot be, then the limitations of the method should be set out clearly.

More specific points:

Some particular queries:

1. There are some areas in the paper where you need to be more precise in writing. For example, you state on line 44 that “The percentage of the population living in urban areas was 58.4% in 1975, 86.6% in 2001, and 91% in 2016.” And on the next line: “This rapid population growth has increased the size of the urban area”    Here you do not differentiate between population growth and redistribution.  To substantiate the 2nd statement fully, what has the overall population growth been, as well as the redistribution between rural and urban areas?

2. Line 53: “The Kharj region was, and still is, one of the attractive areas for the population because of the diversity and multiplicity of resources available at the economic and professional levels.” Will the national urban policies impact on this? How will this region fare compared to the new cities being developed? What are the policies directed towards future development here?

3. Explain how the growth rates are calculated. On line 59: “the urban area in 1974 reached approximately 5.3 km². In 1985, the urban area increased to approximately 12.5 km², i.e., an increase of 74.1%; such increase continued until the area reached 19.5 km² in 2000. In 2010, it reached 39 km², an increase of 50% than it was in the year 2000;”  The percentages set out do not appear to be correct.

4.  In relation to the maps presented, there are new growth areas that have appeared between decades, as opposed to extensions of existing areas. (Eg in figure 4 the area that is marked in orange in the bottom row, 2nd column of the figure.) Can you explain why? Was it related to an urban planning decision, and why, or was it ‘natural’ growth driven by market forces?  Can such new areas of development be predicted?

5. Define and explain the kappa index used for assessing models. 

Author Response

The paper sets out to simulate urban growth in Kharj over the 2020-2040, by modelling it based on the growth observed in the region over the period between 1990 and 2020.  There is nothing mentioned in that paper to suggest that changing circumstances within KSA, and worldwide, are taken into account in setting up the model.

We mentioned that in line(44) (The percentage of the population living in urban areas…) increased, which lead to an increase in  the size of the urban area (worldwide) and in Saudi Arabia which contributed to the urbanization rate, the great demand for land uses in all forms and the increase in an urban area,  because the availability of services and education (line 48)

The authors acknowledge that urban growth is driven by “internal or external migration and an increase in economic activity rates”, but there is no information on where predictions of population growth, birth and migration rates and possible changes in social and economic planning are taken into account.  Given the fast changing circumstances in Saudi society, these would not be expected to remain constant at rates given in the 1990-2020 period.  
Ù…

We added more details in line (385) for the following reference which supports the results of increased built-up area. More Al-Kharj is near Riyadh city which is targeted to reach 15 million.

Populationpyramid.net. 2019. Population Pyramids of the World from 1950 to 2100: WORLD. [online] Available at: <https://www.populationpyramid.net/world/2050/> [Accessed 4 Jun 2022].

Urban growth in terms of land usage would also be determined by planned housing style and density, factors which one would expect will change over time, particularly if the current Saudi policy focus on sustainability within cities and responses to energy concerns, extend beyond the new cities under development.  

Okay, also focus on current cities and how it will be developed is part of the government goals because this research will contribute in same government focus.

The growth forecast in the paper appears to consider only the physical characteristics of the area as determining factors, with the authors listing “the factors affecting urban growth, such as distance from the city center, the road network, valleys, and land slopes, to monitor the prediction of urban growth.”  Social factors (such as future economic activity rates) although briefly acknowledged as important, do not appear to be modelled.  Neither is there any mention of wider social or political policies that will impinge on growth.  It is not realistic to assume these will be unchanged over time.

Yes, we focus on physical characteristics in this research as they can be considered as urban planning issues. Social factors can be considered as future research that uses our results to compare with.

While the model may suggest the likely direction of urban growth in the area over time if the area expands (based on the physical characteristics of the terrain), it seems unrealistic to suggest the speed of growth can be accurately predicted in a context void of the factors highlighted above.

Thanks for the comments, researchers disagreed on this point, this is a common study that will be based on previous historical data.

If the wider social and economic context has been taken into account in modeling, this needs to be made explicit by the authors. If it has not been, or cannot be, then the limitations of the method should be set out clearly.

We mentioned this as a limitation. In line 151 thanks

-----

Some particular queries:

  1. There are some areas in the paper where you need to be more precise in writing. For example, you state on line 44 that “The percentage of the population living in urban areas was 58.4% in 1975, 86.6% in 2001, and 91% in 2016.” And on the next line: “This rapid population growth has increased the size of the urban area”    Here you do not differentiate between population growth and redistribution.  To substantiate the 2ndstatement fully, what has the overall population growth been, as well as the redistribution between rural and urban areas?

In these sentences, we mention that the increment in the population from 1975 to 2016 resulted in to increase the urban areas.

  1. Line 53: “The Kharj region was, and still is, one of the attractive areas for the population because of the diversity and multiplicity of resources available at the economic and professional levels.” Will the national urban policies impact on this? How will this region fare compared to the new cities being developed? What are the policies directed towards future development here?

the new developed cities are far from this region and the target of these cities different than the importance of the alkharj city

  1. Explain how the growth rates are calculated. On line 59: “the urban area in 1974 reached approximately 5.3 km². In 1985, the urban area increased to approximately 12.5 km², i.e., an increase of 74.1%; such increase continued until the area reached 19.5 km² in 2000. In 2010, it reached 39 km², an increase of 50% than it was in the year 2000;”  The percentages set out do not appear to be correct.

Thanks, we corrected it.

  1. In relation to the maps presented, there are new growth areas that have appeared between decades, as opposed to extensions of existing areas. (Eg in figure 4 the area that is marked in orange in the bottom row, 2ndcolumn of the figure.) Can you explain why? Was it related to an urban planning decision, and why, or was it ‘natural’ growth driven by market forces?  Can such new areas of development be predicted?

Thanks we added the explanation in line  220

  1. Define and explain the kappa index used for assessing models. 

We added in line 235

Reviewer 4 Report

I suggest paper publication but some improvements should be made:

Section 4 (Area of the study) could be merged with section 5 (Data and methodology)

In Figure 9 colors should be explained (what represents green and what purple)

Equation on page 12 percentage change in area is not adequate, the formula is representing the proportion of specific area in total area and not change, change should be in different time periods

Why is in Figure 11 background in black? It should be similar as in previous figures

Paper has no conclusion. Therefore alongside recommendations some conclusion should be written covering paper limitations additionally.

Author Response

Reviewer#4, Concern #1 : Section 4 (Area of the study) could be merged with section 5 (Data and methodology).

Author response: 

We thank the reviewer for time spent and valuable feedback.

Author action: We updated the manuscript based on his comment:

Reviewer#4, Concern #2: In Figure 9 colors should be explained (what represents green and what purple)

.

Author response:  We thank the reviewer for time spent and valuable feedback. We updated the manuscript based on his comment.

Author action: … change through a graph of the gains (green) and losses (purple) for the various….

Reviewer#4, Concern #3: Equation on page 12 percentage change in area is not adequate, the formula is representing the proportion of specific area in total area and not change, change should be in different time periods

.

Author response:  We thank the reviewer for time spent and valuable feedback. We updated the manuscript and correct the formula.

Author action:

Reviewer#4, Concern #4: Why is in Figure 11 background in black? It should be similar as in previous figures

Author response:  We thank the reviewer for the time spent and valuable feedback.

Author action: We updated the manuscript by replacing the figure.

Reviewer#4, Concern #5: Paper has no conclusion. Therefore alongside recommendations some conclusion should be written covering paper limitations additionally.

Author response:  We thank the reviewer for the time spent and valuable feedback.

Author action: We updated the manuscript by adding a paragraph and section “Conclusion and Recommendations”

Round 2

Reviewer 2 Report

Thank to authors. I saw changes on literature and some English corrections. However I didn't see any change in the model update by using alternative scenarios to support sustainable urban development. It is important to present not just regular growth (do nothing scenario) but also policy oriented growth simulations. Therefore I prefer to reject this study
Best Regards

Author Response

Author response:  We thank the reviewer for time spent and valuable feedback.

Author action: we agree alternative scenarios help to support sustainable urban development; this will be considered for next research, since the alternative scenarios out of this article scope. this article study the regular growth but the unique about this study is that, we validated the appropriate methodology for this region before we use it, to help regional government and local municipality planning for infrastructure and public services.

Reviewer 3 Report

The authors have not seen relevance in several of my original comments and have not addressed them in the revision.  

The reply states that you addressed the limitation caused by omission of all social and political contexts in forecasting urban growth with a comment in line 151. I do not see social science or political aspects being omitted acknowledged in the text added, though the text does define the realm the model fits in more clearly and alludes to its narrowness. 

There is a sentence in line 174 "Also, the methodology limitations in this research include factors that affect urban growth, such as distance from the city center, road network, valleys and land slopes."  To me this suggests that factors that entered their study are limitations, which I presume is not your intention. 

1. The figures in lines 60-66 still need to be checked. The change from 3.5 square km to 12.5 is written as "an increase of 235%"    (12.5 - 3.5)/3.5 = 2.57  (i..e. 257%)  - is that the form of the calculation being used? 

Then the change from 19.5 square km to 39 is claimed to be "an increase of 200%."  (Consistent with the workings above, the final size for an increase of 200% would have been 58.5km squared.) The size had doubled in the period, it has not experienced an increase of 200%. 

2. Around the the definition of the Kappa Statistic the language of the additional part added needs to be checked, and I would suggest a more scientific definition presented. 

" by Kappa statistical. The Kappa coefficient is the name of the statistician Cohen Kappa, which was developed in 1960 to measure reliability between two raters. It was used in 1980 in remote sensing to express the accuracy of an image classification [24].   

The history of the statistic, and who it is named for, is not relevant; rather the (mathematical) definition of the statistic itself and how to interpret the result is.   (Is the statistic being used Cohen's Kappa? - the statistician is Cohen, not Cohen Kappa.) 

3. You have inserted in line 222 . "Also, as shown in Figure 4, there are new growth areas that have appeared 15km south-west from the intersection of Al Dilam Road and the road connected to Al-Kharj Industrial City. These new growth areas were established in 2011 and are still growing." My query as to whether you can say why the new growth in this area has established (whether due to public planning laws for example, or entirely market driven etc.) was not addressed. 

Author Response

Reviewer#3, Report (Round 2) Concern#1: 1. The figures in lines 60-66 still need to be checked. The change from 3.5 square km to 12.5 is written as "an increase of 235%"    (12.5 - 3.5)/3.5 = 2.57  (i..e. 257%)  - is that the form of the calculation being used?

Then the change from 19.5 square km to 39 is claimed to be "an increase of 200%."  (Consistent with the workings above, the final size for an increase of 200% would have been 58.5km squared.) The size had doubled in the period, it has not experienced an increase of 200%.

Author response: 

We thank the reviewer for time spent and valuable feedback.

Author action: We updated the manuscript based on his comment and correct the figure. thanks

Reviewer#3, Report (Round 2) Concern#2: 2. Around the definition of the Kappa Statistic the language of the additional part added needs to be checked, and I would suggest a more scientific definition presented.

" by Kappa statistical. The Kappa coefficient is the name of the statistician Cohen Kappa, which was developed in 1960 to measure reliability between two raters. It was used in 1980 in remote sensing to express the accuracy of an image classification [24].  

The history of the statistic, and who it is named for, is not relevant; rather the (mathematical) definition of the statistic itself and how to interpret the result is.   (Is the statistic being used Cohen's Kappa? - the statistician is Cohen, not Cohen Kappa.)

Author response: 

We thank the reviewer for time spent and valuable feedback.

Author action: We updated the manuscript based on his comment and correct the name. and added mathematical formula.

Reviewer#3, Report (Round 2) Concern#3: 3. You have inserted in line 222 . "Also, as shown in Figure 4, there are new growth areas that have appeared 15km south-west from the intersection of Al Dilam Road and the road connected to Al-Kharj Industrial City. These new growth areas were established in 2011 and are still growing." My query as to whether you can say why the new growth in this area has established (whether due to public planning laws for example, or entirely market driven etc.) was not addressed.

Author response: 

We thank the reviewer for time spent and valuable feedback.

Author action: We updated the manuscript based on his comment and answer why the new growth was established.